# Robust multiferroic in interfacial modulation synthesized wafer-scale one-unit-cell of chromium sulfide

Luying Song[1,5], Ying Zhao[2,5], Bingqian Xu[3], Ruofan Du[1], Hui Li [1], Wang Feng[1], Junbo Yang[1], Xiaohui Li[1], Zijia Liu[3], Xia Wen[1], Yanan Peng[1], Yuzhu Wang[1], Hang Sun[1], Ling Huang[1], Yulin Jiang[1], Yao Cai[4], Xue Jiang [2], Jianping Shi [1] ✉ & Jun He [3] ✉

Multiferroic materials offer a promising avenue for manipulating digital information by leveraging the cross-coupling between ferroelectric and ferromagnetic orders. Despite the ferroelectricity has been uncovered by ion displacement or interlayer-sliding, one-unit-cell of multiferroic materials design and wafer-scale synthesis have yet to be realized. Here we develop an interface modulated strategy to grow 1-inch one-unit-cell of non-layered chromium sulfide with unidirectional orientation on industry-compatible *c*-plane sapphire. The interfacial interaction between chromium sulfide and substrate induces the intralayer-sliding of self-intercalated chromium atoms and breaks the space reversal symmetry. As a result, robust room-temperature ferroelectricity (retaining more than one month) emerges in one-unit-cell of chromium sulfide with ultrahigh remanent polarization. Besides, long-range ferromagnetic order is discovered with the Curie temperature approaching 200 K, almost two times higher than that of bulk counterpart. In parallel, the magnetoelectric coupling is certified and which makes 1-inch one-unit-cell of chromium sulfide the largest and thinnest multiferroics.

Multiferroic materials are well known for their potential applications in non-volatile memories and sensors, which have been extended to the fields of photovoltaics for efficient renewable energy harvesting and synaptic devices for powerful neuromorphic computing[1–3]. Nevertheless, the existing three-dimensional (3D) multiferroic materials fall short of meeting the industry criteria for practical applications in information storage because of the size limit, interfacial effect, polarization origin, and reversal mechanism[2,4,5]. Two-dimensional (2D) materials have revealed an unprecedented potential to consistently drive advanced device performances due to their unique physical and electronic properties[6–9]. Particularly, 2D ferroelectricity has been uncovered and the spontaneous polarization generally originates from the non-centrosymmetric structure[10–15]. In addition, the weak interlayer interaction in 2D layered materials enables the layer-sliding, which breaks the centrosymmetry and leads to the emergence of polarization[16–23]. Even so, experimental exploration of 2D multiferroic materials still remains a daunting challenge especially at the atomically thin thickness because of the influence of thermal fluctuation and depolarizing field[24]. Furthermore, the relatively small remanent polarization in 2D sliding ferroelectrics limits their practical applications. In this regard, searching for 2D intrinsic multiferroic materials has become the central task for constructing the next-generation information storage devices.

[1]The Institute for Advanced Studies, Wuhan University, Wuhan 430072, China. [2]Key Laboratory of Materials Modification by Laser, Ion and Electron Beams (Ministry of Education), Dalian University of Technology, Dalian 116024, China. [3]Key Laboratory of Artificial Micro- and Nano-Structures of Ministry of Education, School of Physics and Technology, Wuhan University, Wuhan 430072, China. [4]The Institute of Technological Sciences, Wuhan University, 430072 Wuhan, China. [5]These authors contributed equally: Luying Song, Ying Zhao. ✉e-mail: jianpingshi@whu.edu.cn; He-jun@whu.edu.cn

Chromium-based chalcogenides ($Cr_mX_n$, where X = S, Se, and Te) possess tunable structural phases and magnetic orders by means of the stoichiometric variation[25–30]. The alternating stacks of Cr-deficient and Cr-full layers, as well as the self-intercalation of Cr atoms determine such interesting magnetic properties. Interestingly, the intralayer-sliding of self-intercalated Cr atoms may break the space reversal symmetry and induces the spontaneous polarization. Assuming this to be the case, one-unit-cell of $Cr_mX_n$ should be one of the thinnest multiferroic materials. However, the experimental exploration is yet to be realized in view that the strong chemical bond between Cr and X restricts the intralayer-sliding. On the other hand, to meet the industry criteria for practical applications in information storage, the batch synthesis of wafer-scale 2D multiferroic single crystals is a fundamental issue. Despite considerable efforts have been devoted[31–34], the wafer-scale growth of one-unit-cell of non-layered $Cr_mX_n$ still remains challenging because the inherent 3D chemical bond hinders the 2D anisotropic growth.

Here we develop an interface-modulated chemical vapor deposition (CVD) method to synthesize 1-inch one-unit-cell of non-layered chromium sulfide ($Cr_2S_3$) with unidirectional orientation on c-plane sapphire. The relatively strong interfacial interaction is demonstrated between $Cr_2S_3$ and substrate, which determines the unidirectional growth of $Cr_2S_3$ and intralayer-sliding of self-intercalated Cr atoms. As expected, the space reversal symmetry is broken and the room-

temperature ferroelectricity emerges in one-unit-cell of $Cr_2S_3$ featured with ultrahigh stability (retaining more than one month). Meanwhile, the long-range ferromagnetic order is uncovered with the Curie temperature approaching 200 K, almost two times higher than that of bulk counterpart. Combining density functional theory (DFT) calculations and low-temperature quantum transport/piezoresponse force microscopy (PFM) measurements, the internal mechanism of multiferroic is clarified unambiguously. This work provides an innovative strategy for bridging synthesis and multiferroic investigation of wafer-scale one-unit-cell of non-layered $Cr_2S_3$, which enables the construction of multi-terminal spintronic chips and magnetoelectric devices.

## Results

### Synthesis of 1-inch length one-unit-cell of non-layered $Cr_2S_3$

The industry-compatible c-plane sapphire is selected as the substrate because of its atomically flat surface and low surface diffusion barrier of reactants, which contribute to the evolution of atomically thin $Cr_2S_3$. Corresponding rocking curve is shown in Supplementary Fig. 1, where the miscut angles towards A and M axes are measured to be 0° and 0.2°, respectively. Notably, the parallel steps are readily formed on sapphire surfaces at relatively low temperature (980 °C) without any pre-annealing process (Fig. 1a and Supplementary Fig. 2), which is fundamentally important for the nucleation and growth of unidirectionally aligned $Cr_2S_3$[32–35]. Figure 1b reveals the digital

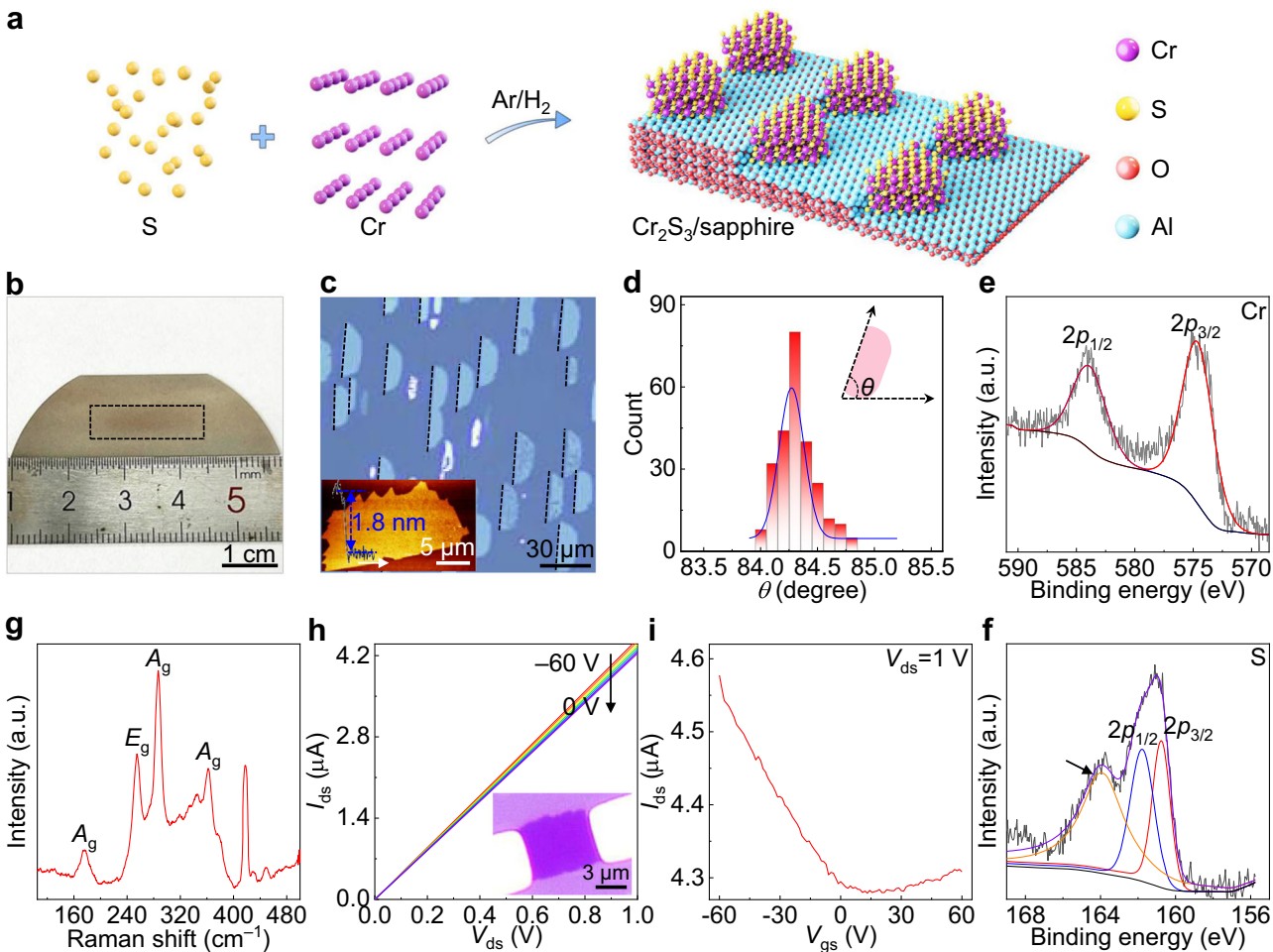

**Fig. 1 | Orientation-controlled synthesize 1-inch length one-unit-cell of non-layered $Cr_2S_3$ on c-plane sapphire. a** Schematic diagram of unidirectionally aligned growth of one-unit-cell of $Cr_2S_3$. **b** Photography of CVD-synthesized 1-inch length $Cr_2S_3$ on c-plane sapphire. **c** OM image of $Cr_2S_3$ on c-plane sapphire, showing the unidirectionally aligned feature. Inset: AFM image and corresponding height profile of a single $Cr_2S_3$ nanosheet, revealing its one-unit-cell nature. **d** Statistical analysis of domain orientations of $Cr_2S_3$ on c-plane sapphire. **e, f** XPS spectra of as-grown samples, showing the formation of $Cr_2S_3$ and Al-S bonds. **g** Raman spectrum of as-grown sample. **h, i** Output and transfer characteristic curves of a $Cr_2S_3$ back-gated FET. Inset: OM image of the device.

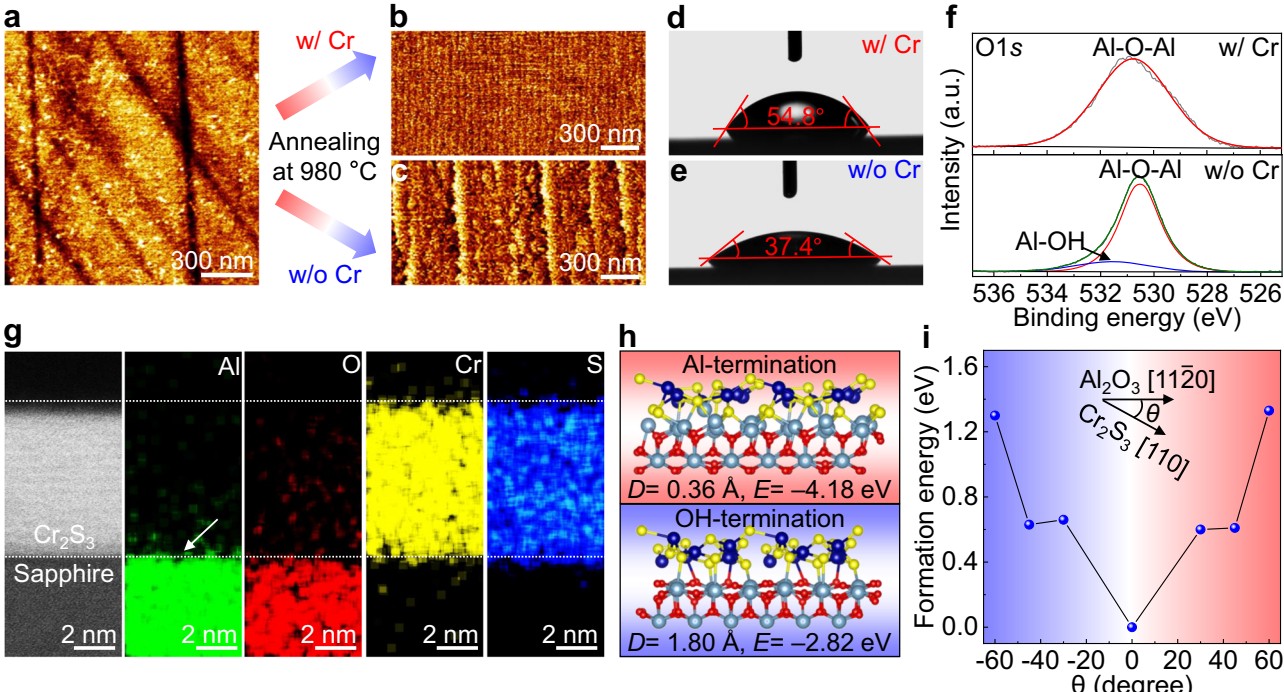

**Fig. 2 | Growth mechanism of unidirectional aligned $Cr_2S_3$ on $c$-plane sapphire.** **a** AFM image of an initial $c$-plane sapphire. **b**, **c** AFM images of $c$-plane sapphire after annealing at 980 °C with and without the presence of Cr powders, respectively. **d**, **e** Contact angles of distilled water droplet on these two distinct sapphire surfaces. **f** O 1$s$ XPS spectra captured on these two distinct sapphire surfaces. **g** Cross-sectional STEM image and corresponding element distributions of as-grown $Cr_2S_3$ on $c$-plane sapphire. **h** Schematic diagrams of $Cr_2S_3$ on these two distinct sapphire surfaces. **i** Formation energy of $Cr_2S_3$ island with different rotation angles on $c$-plane sapphire, where $\theta$ is defined as the angle between [110] direction of $Cr_2S_3$ and [11$\bar{2}$0] direction of $c$-plane sapphire.

photography of 1-inch length $Cr_2S_3$ on $c$-plane sapphire, with the optical microscopy (OM) images shown in Fig. 1c and Supplementary Figs. 3, 4, respectively. The half circular $Cr_2S_3$ nanosheets are observed on $c$-plane sapphire possibly due to the high energy barrier of passing over the step edge of sapphire. The obtained $Cr_2S_3$ nanosheets demonstrate nearly 100% unidirectional alignment feature on $c$-plane sapphire, different from the other 2D materials with random orientations[36]. This is further corroborated by the domain orientation distribution in Fig. 1d. In addition, the carrier gas flow direction reveals negligible influence on the domain orientation of $Cr_2S_3$ (Supplementary Fig. 5), indicating that the $Cr_2S_3$-sapphire interaction and parallel steps dominate the unidirectional aligned growth. Meanwhile, the largest $Cr_2S_3$ nanosheet is obtained with the domain size of 108 μm, as shown in Supplementary Fig. 6. In contrast, small domain-sized triangular or irregular $Cr_2S_3$ nanosheets with random orientations are evolved on Au foil, mica, and highly oriented pyrolytic graphite (HOPG) under the same condition, suggestive of the advantage of sapphire for growing unidirectionally aligned $Cr_2S_3$ (Supplementary Fig. 7). Atomic force microscopy (AFM) image and corresponding height profile analysis in Fig. 1c show that the thickness of $Cr_2S_3$ is 1.8 nm, corresponding to one-unit-cell nature.

To determine the chemical state and elemental composition, X-ray photoelectron spectroscopy (XPS) measurements were performed on as-grown samples, with the results shown in Fig. 1e, f. The binding energies at 574.8 and 584.1 eV are attributed to $Cr^{3+}$, while the peaks of 160.8 and 161.8 eV are assigned to $S^{2-}$, consistent with the XPS results of CVD-synthesized $Cr_2S_3$ on mica[28]. Nevertheless, a new characteristic peak (164.0 eV) is observed in the S 2$p$ XPS spectrum (indicated by the black arrow in Fig. 1f), which derives from the Al-S bonds (Supplementary Fig. 8). Such a result manifests the strong interfacial interaction between $Cr_2S_3$ and sapphire, which determines the unidirectionally aligned growth of $Cr_2S_3$[37]. To further confirm the strong interfacial interaction determining the domain orientation of $Cr_2S_3$,

$Cr_2S_3/WS_2$ vertical heterostructures are synthesized on $c$-plane sapphire and the obtained $Cr_2S_3$ nanosheets possess random orientations (Supplementary Fig. 9). Cross-identification by Raman spectroscopy confirm the formation of $Cr_2S_3$ on $c$-plane sapphire, as shown in Fig. 1g and Supplementary Fig. 10. Notably, the characteristic peaks of $Al_2S_3$ are observed in the Raman spectrum of as-grown one-unit-cell of $Cr_2S_3$ on sapphire, further verifying the formation of Al-S bonds. To clarify the electronic properties of $Cr_2S_3$, a back-gated field-effect transistor (FET) was fabricated using Cr/Au as electrodes. The semiconducting feature with p-type is convinced by the output and transfer characteristic curves in Fig. 1h, i, as well as the temperature-dependent resistance in Supplementary Fig. 11. Briefly, 1-inch length one-unit-cell of non-layered $Cr_2S_3$ semiconductors with unidirectional orientation has been synthesized on industry-compatible $c$-plane sapphire, which provides a strategy for growing wafer-scale single crystals and a platform for constructing high-performance in-memory devices.

## Growth mechanism of unidirectionally aligned $Cr_2S_3$

The interfacial modulation method has been proposed to improve the quality of target materials (e.g. HgCdTe)[38,39], which provides a direction to understand the growth mechanism of well-aligned $Cr_2S_3$ on $c$-plane sapphire. The AFM image of an initial $c$-plane sapphire is shown in Fig. 2a, where no steps is observed. Interestingly, the continuous and parallel steps are evolved on sapphire surfaces after annealing at 980 °C with the presence of Cr powders (Fig. 2b). By contrast, the discontinuous and distorted steps are formed under the same annealing condition without the presence of Cr powders (Fig. 2c). This result suggests that the introduction of Cr promotes the formation of parallel steps on sapphire surfaces, which play a crucial role for synthesizing unidirectionally aligned $Cr_2S_3$. To further expound the surface chemical property discrepancy, the water contact angle measurements were then performed on these two distinct sapphire surfaces. A larger contact angle is obtained on the sapphire surface

(54.8°) after annealing with the presence of Cr powders than that on the counterpart (37.4°) after annealing without the presence of Cr powders (Fig. 2d, e and Supplementary Fig. 12), possibly due to the variation of sapphire surface-terminated structure.

In addition, the chemical state of sapphire surface was also detected by XPS, with the results shown in Fig. 2f. Two characteristic peaks corresponding to Al-OH (531.5 eV) and Al-O-Al (530.5 eV) are observed in the O 1$s$ spectrum, which is obtained on the sapphire surface after annealing without the presence of Cr powders. This is totally different from the sapphire surface after annealing with the presence of Cr powders, where only one characteristic peak (530.5 eV) is observed. Such results indicate that the sapphire surface structure is changed from OH- to Al-termination after annealing with the presence of Cr powders, and the similar phenomenon is also demonstrated for high-temperature pre-annealing process[40]. Meanwhile, the formation of Al-S bonds further verifies the Al-terminated surface structure of sapphire. The cross-sectional scanning transmission electron microscopy (STEM) and corresponding energy-dispersive spectroscopy (EDS) characterization results reveal that Al elements are assembled at the interface between $Cr_2S_3$ and sapphire, as shown in Fig. 2g. The atomic-resolution cross-sectional STEM image in Supplementary Fig. 13 further verifies the formation of Al-terminated structure. DFT calculations were then carried out to clarify the growth mechanism of $Cr_2S_3$ on $c$-plane sapphire (Fig. 2h, i and Supplementary Fig. 14). A smaller $Cr_2S_3$-sapphire distance ($D$) is achieved on the Al-terminated surface (0.36 Å) than that on the OH-terminated surface (1.80 Å). Meanwhile, the adsorption energy ($E$) of $Cr_2S_3$ on Al-terminated surface is calculated to be −4.18 eV, much higher than that on OH-terminated surface (−2.82 eV). The small $Cr_2S_3$-sapphire distance and high adsorption energy manifest the strong interfacial interaction between $Cr_2S_3$ and sapphire, which contributes to the unidirectionally aligned growth of $Cr_2S_3$. Besides, the formation energies of $Cr_2S_3$ island with different rotation angles on $c$-plane sapphire were also calculated (Fig. 2i), demonstrating the crucial role of parallel steps on $c$-plane sapphire surfaces for the epitaxial growth of one-unit-cell thick $Cr_2S_3$ with unidirectional orientation. Particularly, unidirectionally aligned $Cr_2Se_3$ nanosheets have also been synthesized on $c$-plane sapphire using the same approach, indicating the universality of this interface-modulated growth strategy (Supplementary Fig. 15).

**Multiscale determining unidirectional alignment and seamless stitching of $Cr_2S_3$**

To explore the atomic structure and unidirectional alignment of CVD-synthesized one-unit-cell of $Cr_2S_3$, multiscale characterizations were thus performed on as-grown and transferred samples. Figure 3a shows the low-magnification TEM image of a transferred $Cr_2S_3$ nanosheet, the well-defined morphology and uniform color contrast indicate its high crystalline quality and thickness uniformity. The corresponding selected-area electron diffusion (SAED) pattern reveals only one set of spots, suggestive of its single-crystal feature (Fig. 3b and Supplementary Fig. 16). As exhibited by the atomic-resolution TEM image in Fig. 3c, a perfect honeycomb lattice is observed to show almost no visible defect, indicating the high crystalline quality. The (110) lattice plane spacing is calculated to be 0.30 nm, consistent with CVD-synthesized $Cr_2S_3$ nanosheets on mica[28]. To further identify the element constitutions and their distributions, EDS characterizations were then performed, with the results shown in Fig. 3d–g and Supplementary Fig. 17. The uniform color contrast within the nanosheet manifests the high crystalline quality of CVD-derived $Cr_2S_3$. The atomic ratio of Cr to S revealed by quantified elemental analysis is calculated to be 38.57:61.43 ≈ 2:3, demonstrating the perfect stoichiometric ratio of $Cr_2S_3$.

A series of SAED patterns captured from different nanosheets reveal nearly the identical orientations, confirming the unidirectional alignment of $Cr_2S_3$ (Fig. 3h, i and Supplementary Fig. 18). To realize the

wafer-scale synthesis of single crystal, the seamless stitching of unidirectional aligned nanosheets is crucial[32,35,41]. The polarized second-harmonic generation (SHG) spectra captured from two merged $Cr_2S_3$ domains reveal nearly the identical patterns, indicating the same orientations (Fig. 3j). Meanwhile, the corresponding SHG mapping reveals no obvious intensity drop across the grain boundary, confirming the seamless stitching of such two $Cr_2S_3$ domains (Fig. 3k). In contrast, the SHG spectra measured on two misoriented $Cr_2S_3$ domains exhibit a twist angle and a sharp boundary (Supplementary Fig. 19a, b). To further verify the seamless stitching of unidirectional aligned $Cr_2S_3$, the $Ar/O_2$ etching experiments were then performed, and the grain boundary was obviously observed for the two merged $Cr_2S_3$ domains with random orientations (Supplementary Fig. 19c, d). Nevertheless, for the unidirectionally aligned $Cr_2S_3$, almost no contrast variation is presented, indicating their seamless stitching, as shown in Fig. 3l, m. Besides, the reciprocal space mapping (RSM) characterizations were also conducted on as-grown samples to convince the unidirectionally aligned feature of $Cr_2S_3$ at millimeter scale (Fig. 3n, o). The single diffraction pattern and the same direction between $Cr_2S_3$ (012) and $Al_2O_3$ (006) suggest the epitaxial growth mode of unidirectionally aligned $Cr_2S_3$ on $c$-plane sapphire (Fig. 3p). In brief, the achievement of 1-inch length one-unit-cell of $Cr_2S_3$ with unidirectional orientation and their seamless stitching provide the cornerstone for synthesizing wafer-scale 2D single crystals.

**Room-temperature ferroelectricity in one-unit-cell of $Cr_2S_3$**

The symmetry of $Cr_2S_3$ (with the thickness changing from 1.9 to 12.3 nm) was investigated by nonlinear optical SHG measurements (Supplementary Fig. 20). A prominent characteristic peak is observed at 532 nm (corresponding to the half of excitation wavelength) for all the testing samples, indicating the non-centrosymmetric structure of $Cr_2S_3$. The unusual space reversal symmetry broken possibly induces the emergence of ferroelectricity and then PFM measurements are performed on transferred samples onto Au/Si, with the schematic diagram shown in Supplementary Fig. 21. Notably, in view of the strong interfacial interaction between $Cr_2S_3$ and sapphire surface, the polypropylene carbonate (PPC) with high viscosity is selected as the supporting layer and the detailed etching-free transfer process is described in Supplementary Fig. 22. The PFM phase image after a box-in-box writing with a tip bias of positive and negative voltages (8 V) shows a well-defined region of phase contrast, corresponding to the remanent polarization (Fig. 4a). This result indicates that the polarization state can be rewritten, highlighting the switchable polarization feature of ultrathin $Cr_2S_3$ especially down to one-unit-cell thickness, which is also confirmed by the amplitude image in Fig. 4b. In addition, the butterfly loops of amplitude signal and the distinct 180° switching of phase corroborate the ferroelectric polarization in one-unit-cell of $Cr_2S_3$ (Fig. 4c, d). Notably, the interfacial charge and internal electric field induced by the distinct electrodes (PFM tip and Au) result in the asymmetry of PFM phase and amplitude hysteresis loops. To the best of our knowledge, it is the first report about the ferroelectricity of $Cr_mX_n$. All the testing $Cr_2S_3$ nanosheets with different thicknesses show the room-temperature ferroelectricity, nevertheless, the polarization signal gradually weakens with increasing the thickness (Supplementary Fig. 23). As the thickness reaches to 73.9 nm, no hysteresis loop or butterfly-like amplitude curve is observed (Supplementary Fig. 24).

To further verify the ferroelectricity in CVD-derived 2D $Cr_2S_3$, the macroscopic ferroelectric hysteresis loop was measured, and the remanent polarization value of one-unit-cell of $Cr_2S_3$ was calculated to be 0.80 μC/cm² (Fig. 4e). Particularly, the ferroelectricity is still maintained even after one month and the remanent polarization shows negligible variation, as revealed in Fig. 4e, indicating the robust room-temperature ferroelectricity in 2D $Cr_2S_3$. The remanent polarization value as large as 4.30 μC/cm² is observed for the $Cr_2S_3$ nanosheet with the thickness of 13.0 nm (Fig. 4f), which is higher than

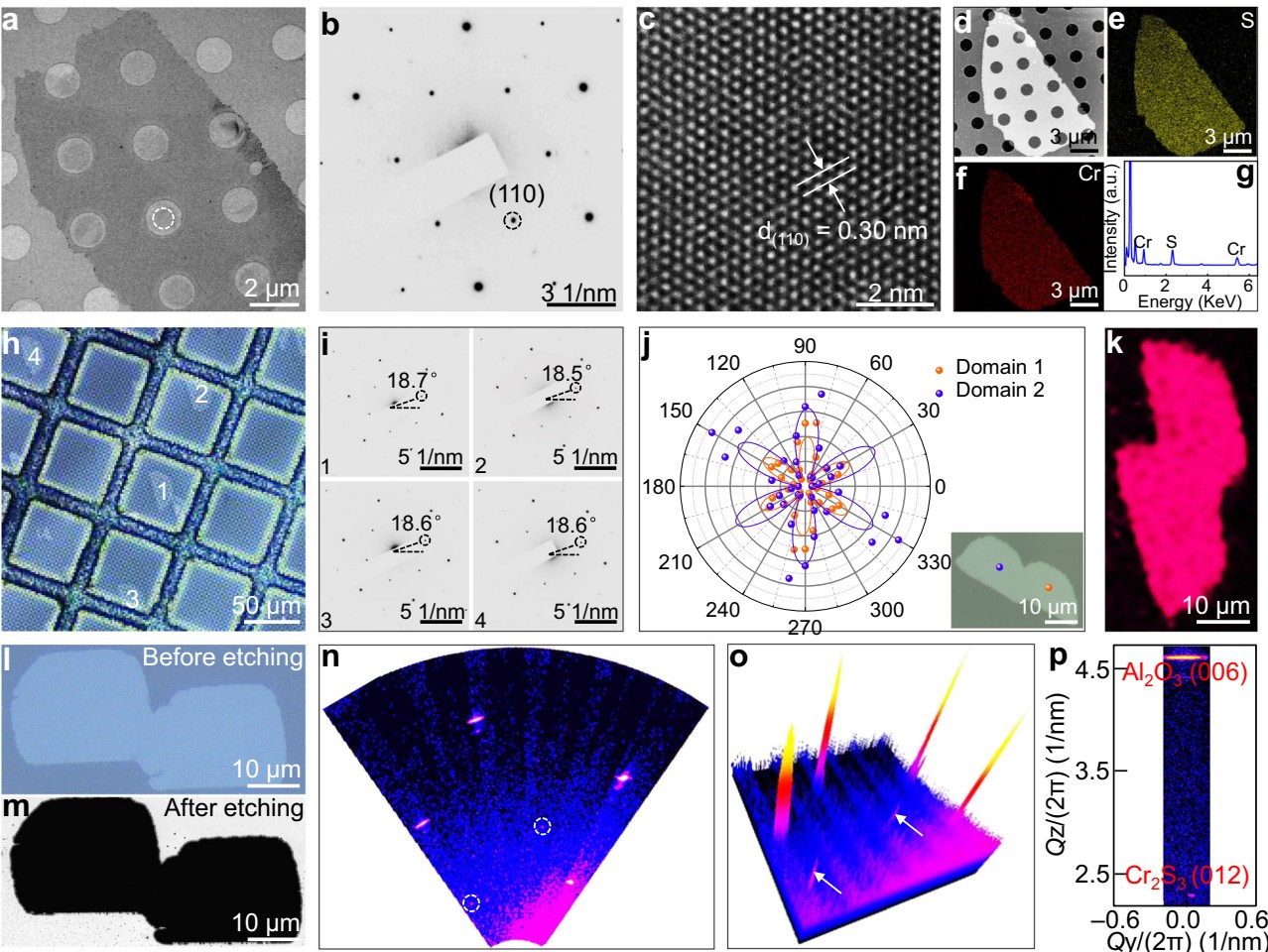

**Fig. 3 | Unidirectional alignment and seamless stitching of Cr$_2$S$_3$. a** Low-magnification TEM image of a transferred Cr$_2$S$_3$ nanosheet. **b** Corresponding SAED pattern captured from the circled region in (**a**), indicating its single-crystal feature. **c** Atomic-resolution TEM image of a transferred Cr$_2$S$_3$ nanosheet, showing its high crystalline quality. **d** Low-magnification STEM image of a transferred Cr$_2$S$_3$ nanosheet. **e**, **f** Corresponding EDS mapping images of Cr and S, revealing the uniform element distributions. **g** Quantified elemental analysis of Cr$_2$S$_3$. **h** OM image of transferred Cr$_2$S$_3$ nanosheets. **i** Corresponding SAED patterns captured from different Cr$_2$S$_3$ nanosheets, showing nearly the identical orientations. **j**, **k** Polarized SHG spectra and mapping image measured on two merged Cr$_2$S$_3$ domains with unidirectional alignment. Inset: OM image of such two merged domains. **l**, **m** OM and scanning electron microscopy images of two merged Cr$_2$S$_3$ domains with unidirectional alignment before and after Ar/O$_2$ etching. **n–p** RSM results of as-grown Cr$_2$S$_3$ on sapphire, highlighting the unidirectional alignment of Cr$_2$S$_3$ at millimeter scale.

other 2D ferroelectric materials (Table 1). Besides of the interfacial interaction, the ionic polar displacement and interlayer charge transfer possibly contribute to enhance the polarization of thick Cr$_2$S$_3$, and related theoretical explorations are expected to be made in the future. Furthermore, the cumulative effect of self-intercalated Cr atoms and CrS$_2$ layers sliding, as well as the weakened depolarization field should also result in the polarization elevation, as have been demonstrated in other ferroelectric materials[42,43]. In addition, the macroscopic ferroelectric hysteresis loop measurement of bulk Cr$_2$S$_3$ is performed, with the result shown in Supplementary Fig. 25. The counterclockwise hysteresis loops of transfer characteristic curves at different sweep rates and drain-source voltages are obtained in the vertical FET, and the large hysteresis window (40 V) further verifies the ferroelectric polarization in Cr$_2$S$_3$ (Supplementary Fig. 26).

To uncover the origin of ferroelectricity in 2D Cr$_2$S$_3$, the atomic-resolution cross-sectional STEM measurements were performed, with the results shown in Fig. 4g–j. Interestingly, unexpected intralayer-sliding of self-intercalated Cr atoms is observed, especially at the Cr$_2$S$_3$/sapphire interface (Fig. 4h). Away from the interface, the intralayer-sliding value of self-intercalated Cr atoms is reduced accordingly (from 0.6 to 0.3 Å), which should result in the

attenuation of ferroelectricity. The strong interfacial interaction between Cr$_2$S$_3$ and sapphire surface induces such an unusual intralayer-sliding of self-intercalated Cr atoms. The similar phenomena have also been demonstrated in monolayer GaSe[44] and γ-InSe[45]. Besides, DFT calculations were carried out to further understand the internal mechanism of ferroelectricity in 2D Cr$_2$S$_3$ (Fig. 4k, l). Comparing with the pristine AAA stacking of Cr$_2$S$_3$ (Supplementary Fig. 27a), the strong interfacial interaction between Cr$_2$S$_3$ and sapphire surface induces the self-intercalated Cr atoms sliding and a new ABA stacking order is built. The intercalated Cr atom in the upper interlayer forms a distorted trigonal prismatic coordination with S atoms, while the intercalated Cr atom in the lower interlayer forms a distorted trigonal antiprismatic coordination with S atoms (Fig. 4l). During the sliding of central CrS$_2$ layer, the coordination environment of self-intercalated Cr atoms changes accordingly. The final state is obtained as the upper self-intercalated Cr atoms form a distorted trigonal antiprismatic coordination with S atoms and the lower self-intercalated Cr atoms form a distorted trigonal prismatic coordination with S atoms. The plane-averaged charge densities along $z$ direction are plotted in Supplementary Fig. 27b to clarify the charge distribution of two polarization states.

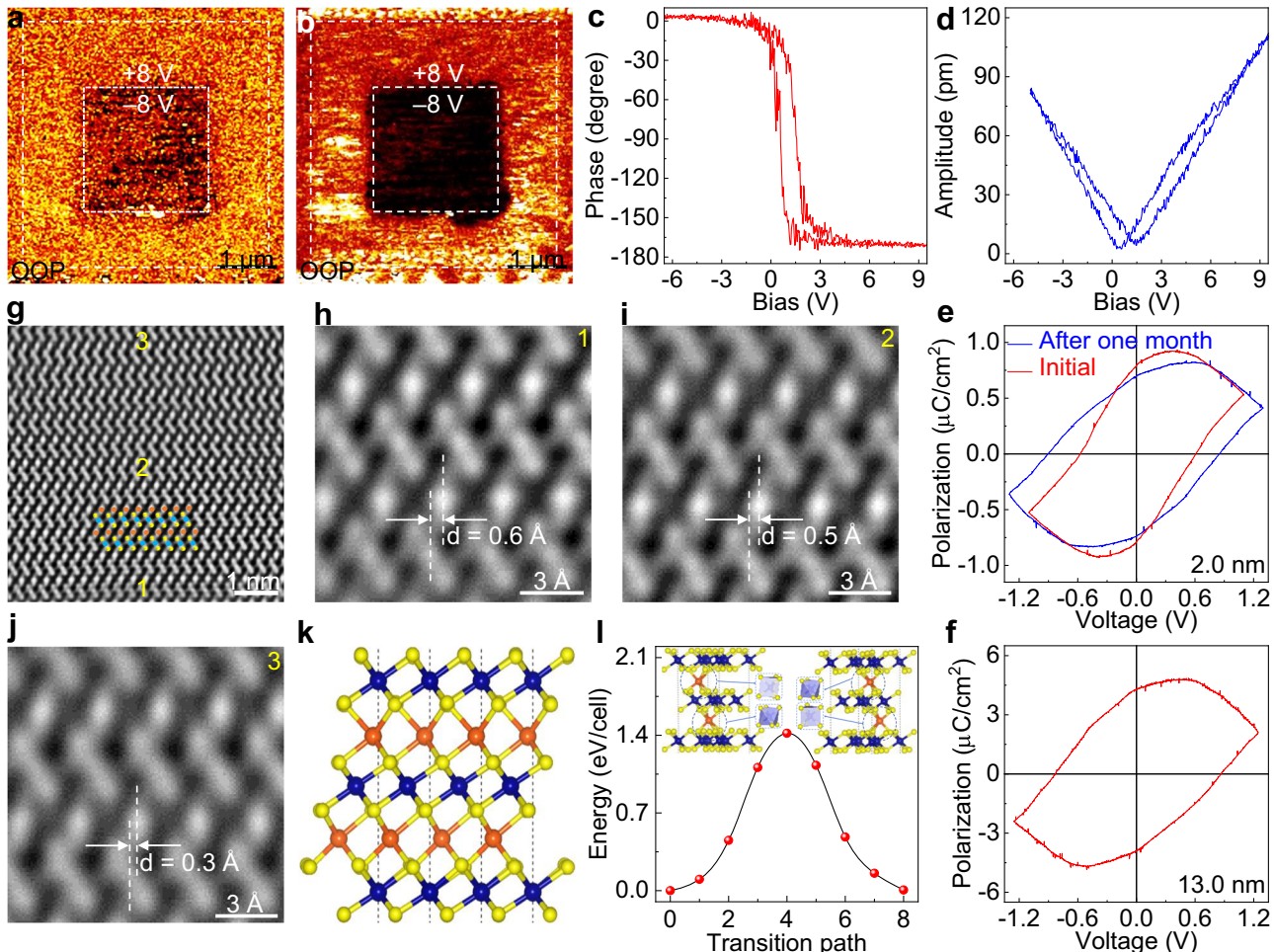

**Fig. 4 | Room-temperature ferroelectricity in CVD-synthesized one-unit-cell of non-layered Cr₂S₃. a, b** PFM phase and amplitude images of one-unit-cell of $Cr_2S_3$ after poling with ±8 V, indicating its stable polarization state. **c, d** Corresponding PFM phase and amplitude hysteresis loops of one-unit-cell of $Cr_2S_3$.
**e, f** Macroscopic polarization hysteresis loops of $Cr_2S_3$ with the thicknesses of 2.0 nm and 13.0 nm, respectively, showing its robust and intrinsic room-temperature ferroelectricity. **g–j** Cross-sectional STEM images of as-grown $Cr_2S_3$ on *c*-plane sapphire, displaying the intralayer-sliding of self-intercalated Cr atoms.

**k** Atomic structure of one-unit-cell of $Cr_2S_3$ with intralayer-sliding of self-intercalated Cr atoms. The blue, yellow, and orange spheres represent the Cr, S atoms in $CrS_2$ layers and the self-intercalated Cr atoms, respectively. **l** Energy evolution between the two opposite polarization states of one-unit-cell of $Cr_2S_3$. The energy decreasing from the centrosymmetric to non-centrosymmetric structure indicates a continuous and spontaneous phase transition between these two phases.

**Table 1 | The remanent polarization and stability comparison for Cr₂S₃ with other 2D ferroelectric materials**

| Material | Thickness | Remanent polarization | Stability | Method | References |
|---|---|---|---|---|---|
| 1T′-ReS₂ | Multilayer | 0.68 pC/m | / | DFT | 54 |
| Graphene/BN | Bilayer | 0.18 μC/cm² | / | Device | 17 |
| BN | Bilayer | 0.68 μC/cm² | / | Device | 18 |
| VS₂ | Bilayer | 0.202 μC/cm² | / | DFT | 55 |
| MoS₂/WS₂ | Bilayer | 1.45 pC/m | / | Device | 20 |
| WSe₂ | Trilayer | 0.53 pC/m | / | KPFM | 16 |
| Cr₂S₃ | 2.0 nm | 0.80 μC/cm² | One month | Device | This work |
|  | 13.0 nm | 4.30 μC/cm² |  |  |  |

For the initial state, the charge near the lower intercalated Cr atom is more than that of the upper intercalated Cr atom, whereas the final state is opposite. The net charge between upper and lower intercalated Cr atoms results in the interfacial charge transfer and induces the spontaneous polarization. The energy difference between polar and non-polar structure is calculated to be 1.4 eV/cell (comparable to that of $Bi_6O_9$ film[46]), implying the relatively high stability of polar phase. In addition, the remanent polarization value

of one-unit-cell thick $Cr_2S_3$ is calculated to be 0.10 μC/cm², consistent with the experimental result (Supplementary Fig. 27c).

### Ferromagnetism in 2D non-layered Cr₂S₃

2D magnetic materials have shown an unprecedented potential for constructing high-performance non-volatile spintronic memory devices[7,47,48]. Nevertheless, the relatively low Curie temperature and inferior environmental stability fall short of meeting the criteria for

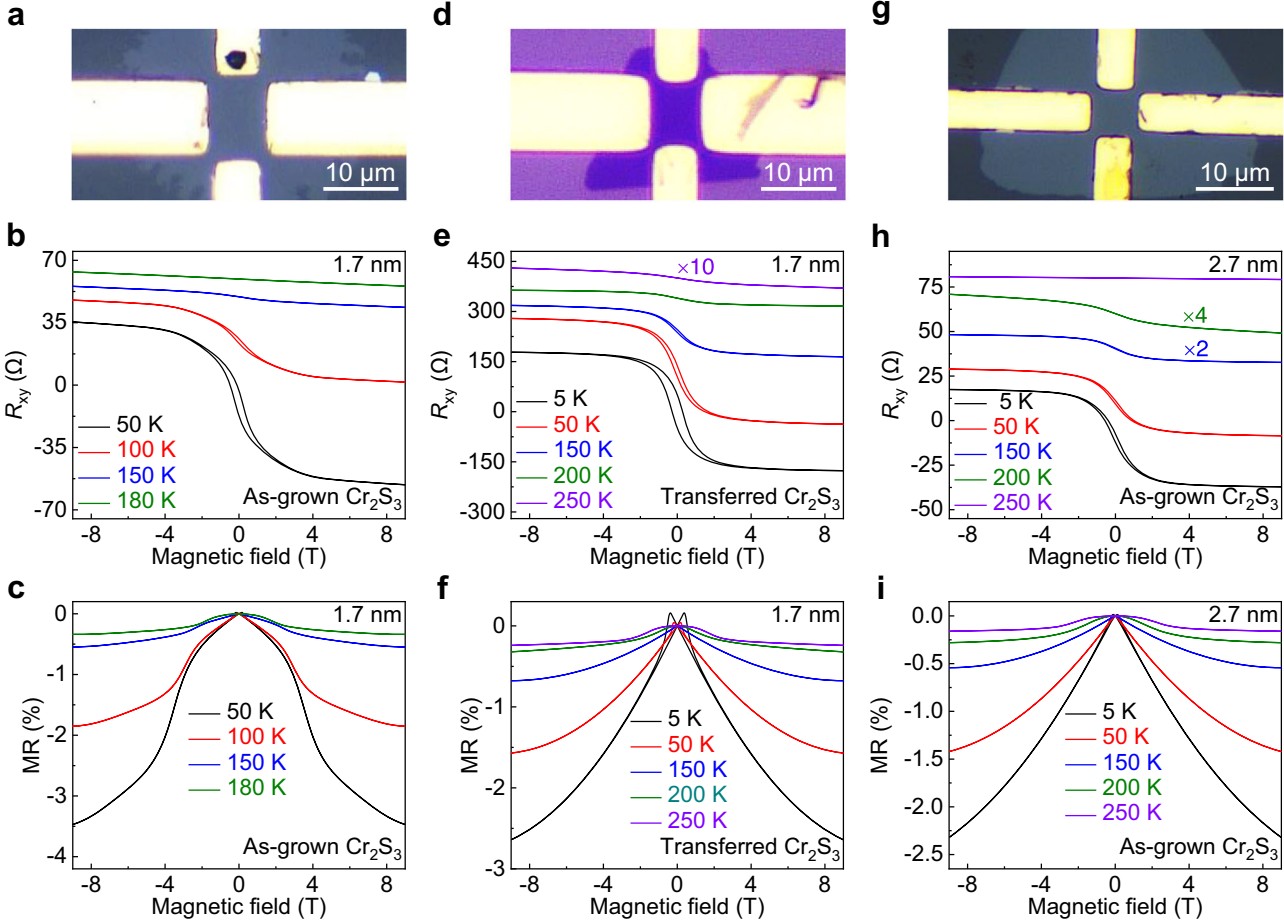

**Fig. 5 | Ferromagnetism in CVD-synthesized one-unit-cell of non-layered $Cr_2S_3$.** **a**, **d** OM images of Hall-bar devices for as-grown and transferred one-unit-cell of $Cr_2S_3$ nanosheets. **b**, **e** Anomalous Hall effects of as-grown and transferred one-unit-cell of $Cr_2S_3$ at various temperatures. The hysteresis below critical temperature indicates the ferromagnetism. **c**, **f** MRs of as-grown and transferred one-unit-cell of $Cr_2S_3$ at various temperatures. **g** OM image of Hall-bar device for an as-grown $Cr_2S_3$ nanosheet with the thickness of 2.7 nm. **h**, **i** Corresponding anomalous Hall effect and MR at various temperatures.

practical applications. The abundant phase states and unique self-intercalation of Cr atoms in $Cr_mX_n$ provide a new paradigm for modulating ferromagnetism. In this regard, low-temperature quantum transport measurements were performed to investigate the magnetism of one-unit-cell thick $Cr_2S_3$ and the OM image of Hall-bar device is displayed in Fig. 5a. As shown in Fig. 5b, as the magnetic field below critical values, the Hall resistances ($R_{xy}$) increase with decreasing the magnetic field. And then, when the magnetic fields are above critical values, the anomalous Hall effects evidenced by the saturated Hall resistance plateau are observed up to 150 K, indicating the spontaneous magnetization and long-range ferromagnetic order in one-unit-cell of $Cr_2S_3$. Moreover, the negative magnetoresistances (MRs) are also detected below 150 K, confirming the emergence of ferromagnetism (Fig. 5c).

To eliminate the interfacial effect between $Cr_2S_3$ and sapphire surface, the magneto-transport properties of a transferred $Cr_2S_3$ nanosheet onto $SiO_2$/Si were then explored (Fig. 5d). The remarkable anomalous Hall effects and negative MRs are unambiguously detected, which suggests the robust ferromagnetism in $Cr_2S_3$ even down to one-unit-cell thickness (Fig. 5e, f). In addition, the influence of thickness on ferromagnetism was also clarified. After increasing the thickness to 2.7 nm, the Curie temperature increases to 200 K, almost two times higher than that of bulk counterpart[28], the intralayer-sliding of self-intercalated Cr atoms should result in such an interesting phenomenon (Fig. 5g–i and Supplementary Figs. 28, 29). The comparison of Curie temperature of $Cr_2S_3$ with other $Cr_mX_n$ is shown in Table 2.

Besides, the environmental stability is crucial for exploring ferromagnetism and constructing multifunctional device, especially at one-unit-cell thickness, and the multiscale characterization results indicate the robust stability of CVD-synthesized 2D $Cr_2S_3$, as shown in Supplementary Fig. 30.

## Magnetoelectric coupling in 2D non-layered $Cr_2S_3$

Electric field-assisted magnetic force microscopy (MFM) is a non-destructive technology for determining the magnetoelectric coupling in bulk[49] and 2D multiferroics[50], and which is performed on as-grown $Cr_2S_3$ with different thicknesses (Fig. 6a, i and Supplementary Fig. 31). After applying a positive voltage (+4 V), the phase deviation between $Cr_2S_3$ and non-magnetic sapphire substrate is detected to be 0.034°, nevertheless, as the voltage changes to −4 V, the phase deviation tunes to be −0.045° (Fig. 6b, c). Further increasing the voltages to ±6 and ±8 V, these phenomena are also observed and large phase deviations are obtained (e.g. 0.056° and −0.054° for +6 and −6 V, 0.109° and −0.092° for +8 and −8 V), as shown in Fig. 6d–g. To further uncover the modulation of electric field on magnetism, the voltage-dependent phase deviations are collected and revealed in Fig. 6h. The variation of phase deviation from positive to negative values is due to the spin flip after applying the electric field with different directions. Several results self-consistently demonstrate the robust magnetoelectric coupling in 2D $Cr_2S_3$ even down to one-unit-cell thick (Fig. 6i–k). The intralayer-sliding of self-intercalated Cr atoms enhances the superexchange interaction of Cr at different positions, which increases the

**Table 2 | The comparison of Curie temperature of Cr₂S₃ with other Cr$_m$X$_n$**

| Material | Substrate | Thickness | Curie temperature | Method | References |
|---|---|---|---|---|---|
| CrSe₂ | WSe₂ | 10.8 nm | 110 K | Reflective magnetic circular dichroism | 25 |
| | | 0.7 nm | 65 K | | |
| trigonal Cr₅Te₈ | SiO₂/Si | 6.0 nm | 125 K | Reflective magnetic circular dichroism | 26 |
| monoclinic Cr₅Te₈ | | | 150 K | | |
| CrTe₂ | SiO₂/Si | 40.0 nm | 179 K | Reflective magnetic circular dichroism | 27 |
| | | 3.0 nm | 189 K | | |
| Cr₂S₃ | Mica | Bulk | 120 K | Vibrating sample magnetometer | 28 |
| Cr₂S₃ | c-plane sapphire | 1.7 nm | 150 K | Anomalous Hall effect | This work |
| | | 2.7 nm | 200 K | | |

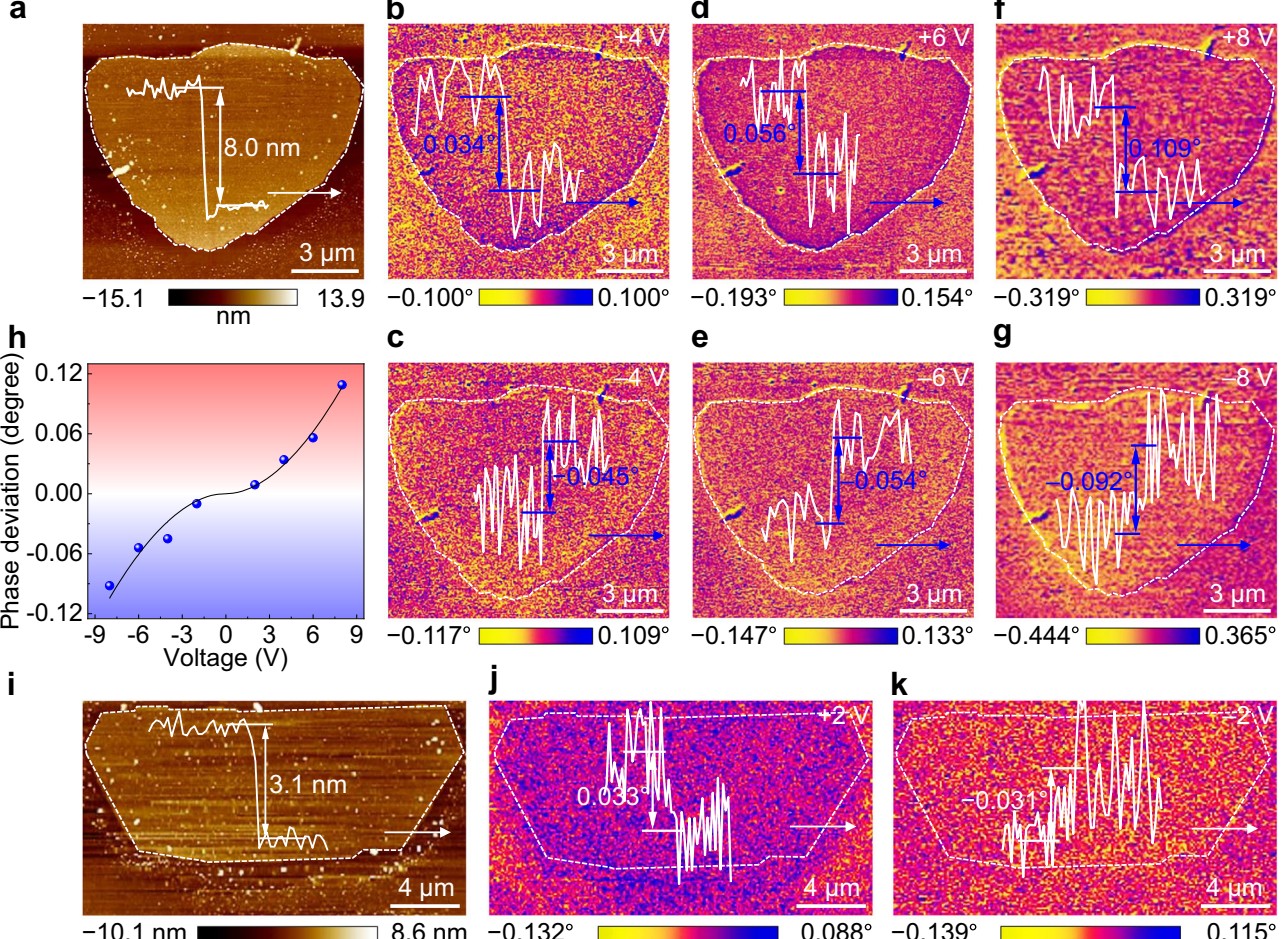

**Fig. 6 | Robust magnetoelectric coupling in 2D non-layered Cr₂S₃. a** AFM image and height profile of a single Cr₂S₃ nanosheet with the thickness of 8.0 nm. **b**–**g** MFM phase images of the Cr₂S₃ nanosheet, after applying the voltages of ±4 V, ±6 V, and ±8 V, respectively. **h** Voltage-dependent phase deviation of the Cr₂S₃ nanosheet. **i** AFM image and height profile of a single Cr₂S₃ nanosheet with the thickness of 3.1 nm. **j**, **k** MFM phase images of the Cr₂S₃ nanosheet, after applying the voltages of ±2 V, respectively.

magnetocrystalline energy and Curie temperature of Cr₂S₃. Meanwhile, the intralayer-sliding of self-intercalated Cr atoms breaks the space reversal symmetry and induces the *p-d* orbital hybridization between S and Cr ions, which result in the emergence of room-temperature ferroelectricity and magnetoelectric coupling in 2D Cr₂S₃.

## Discussion

In summary, we design a facile CVD approach to synthesize 1-inch one-unit-cell of non-layered Cr₂S₃ with unidirectional orientation on c-plane sapphire. The introduction of Cr changes the terminated

structure of sapphire surface, increases the interfacial interaction between Cr₂S₃ and substrate, induces the parallel steps formation on sapphire surface at low temperature, which contribute to the edge-nucleation of unidirectionally aligned Cr₂S₃ nanosheets and single crystal synthesis. Particularly, the interaction between Cr₂S₃ and sapphire surface results in the intralayer-sliding of self-intercalated Cr atoms, breaks the space reversal symmetry, and then the room-temperature ferroelectricity emerges and even maintains more than one month. The long-range ferromagnetic order is also obtained with the Curie temperature approaching 200 K. The coexistence and robust

coupling of ferroelectricity and ferromagnetism make 1-inch one-unit-cell of non-layered $Cr_2S_3$ the largest and thinnest multiferroic materials. These results present a breakthrough toward the batch synthesis of wafer-scale one-unit-cell of multiferroic single crystals, and open up an avenue for future industrial implementation of multiferroic materials in the next-generation magnetoelectric devices.

## Methods

### Synthesis of unidirectionally aligned $Cr_2S_3$

Before the CVD growth, the c-plane sapphire substrates (purchased from Nanjing MKNANO Tech. Co., Ltd.) were cleaned by detergent, deionized water, and ethanol, respectively. The growth of 1-inch one-unit-cell of non-layered $Cr_2S_3$ with unidirectional orientation was conducted in a dual-heating-zone furnace. The sulfur (99.5%, thermo scientific) and Cr powders (99.5%, Innochem) mixed with a few of potassium iodide (KI) particles were selected as the precursors. Notably, the introduction of KI should reduce the evaporating temperature of Cr powders significantly. Before heating, 500 standard cubic centimeters (sccm) argon (Ar) was purged into the chamber for 10 minutes to remove the residual air and humidity. Subsequently, the first and second zones were heated to 170 and 980 °C, respectively, with 110 sccm Ar and 10 sccm hydrogen ($H_2$) as the carrier gases. The growth time was set to be 25 min. After completing the CVD growth process, the furnace cover was opened and cooled down to room-temperature.

### Etching-free transfer process

CVD-synthesized one-unit-cell of $Cr_2S_3$ nanosheets were transferred onto target substrates via a PPC-assisted method. In detail, the PPC solution was prepared by dissolving 1 g PPC particles in 5 mL anisole and then was spin-coated onto $Cr_2S_3$/sapphire for 50 s at 2000 rpm/min. After baking the PPC/$Cr_2S_3$/sapphire for 10 min at 90 °C, the edges were scraped for the fast separation of PPC/$Cr_2S_3$ from sapphire surface. The target substrates were used to pick up PPC/$Cr_2S_3$ and then dried at 110 °C. Finally, the PPC/$Cr_2S_3$ was soaked in acetone to remove the PPC supporting layer.

### Multiscale characterizations

The morphology, domain size, thickness, chemical component, and crystalline quality of CVD-synthesized 2D $Cr_2S_3$ were systematically characterized by OM (Olympus BX53M), AFM (Dimension Icon, Bruker), XPS (ESCALAB 250Xi, Mg Kα as the excitation source), SEM (Hitachi S-4800 with the acceleration voltage of 5 KV), Raman spectroscopy (XploRA plus, Horiba with the excitation light of 532 nm), and TEM (JEOL JEM-F200 and JEM-NEOARM with the acceleration voltage of 200 kV). The atomic resolution HAADF-STEM and EDS results were obtained from a spherical-aberration-corrected STEM JEM-ARM200CF with an acceleration voltage of 200 kV. XRD and RSM measurements were performed using a Philips X' Pert diffractometer.

### Electrical and magneto-transport measurements

The CVD-synthesized $Cr_2S_3$ nanosheets were firstly transferred onto $SiO_2$/Si via PPC-assisted method. The devices were fabricated using an ultraviolet maskless lithography machine (TuoTuo Technology (Suzhou) Co., Ltd.). The thermal evaporation system was employed for depositing Cr/Au electrodes with the thicknesses of 5/70 nm. The electrical transport measurements were performed under the vacuum (<1.3 mTorr) and dark conditions using a semiconductor characterization system (Keithley 4200-SCS). Low-temperature quantum transport measurements of $Cr_2S_3$ were conducted in a 9T-Physical Property Measurement System (PPMS, Quantum Design, Dynacool) by constructing a four-terminated Hall bar device. The Hall resistances were measured with the perpendicular magnetic field up to 9 T, and the testing temperature range was set to be 2–250 K with a current of 10 μA.

### Ferroelectric characterizations and magnetoelectric coupling measurements

PFM measurements were carried out under the ambient condition using a constant mode AFM (Bruker Multimode 8) equipped with a Pt/Ir-coated Si cantilever tip (spring constant: 3 N/m). For the local electrical measurements, the bias of ±8 V was applied to the samples. The macroscopic ferroelectric hysteresis loops were measured using the Multiferroic II precision materials analyzer (Radiant Technologies) in a dark box at room-temperature. The electric field-assisted MFM measurements (Dimension Icon with a magnetic CoCr-coated tip in AFM tapping mode, Bruker) were performed on as-grown $Cr_2S_3$ to determine the magnetoelectric coupling with the voltages of 0 - ±8 V.

### DFT calculations

All DFT calculations in this work were performed using the Vienna Ab initio Simulation Package (VASP) with projector augmented wave (PAW) potentials[51]. The generalized gradient approximation (GGA) proposed by Perdew, Burke, and Ernzerhof was selected for calculating the exchange-correlation potential[52]. The cut-off energy for plane wave expansion was set to be 520 eV. The energy criterion was set to be $10^{-5}$ eV in iterative solution of the Kohn-Sham equation. A vacuum layer of 15 Å was added perpendicular to the nanosheet to avoid the artificial interaction between periodic images. The Brillouin zone integration was performed using a $3 \times 3 \times 1$ Monkhorst-Pack k-point mesh. All the geometric structures were fully relaxed until the force was below 0.01 eV/Å.

### Ferroelectric calculations

The DFT calculations were performed using the VASP[51]. The ion-electron interactions were interpreted using the PAW approach[53] and the GGA expressed by the Perdew-Burke-Ernzerhof (PBE) functional was used to describe the exchange-correlation effects[52]. The plane-wave cutoff energy of 500 eV was employed for all the calculations. The convergence criteria were set to be $10^{-6}$ eV in energy and 0.01 eV/Å in force. The Brillouin zone integration was performed using a $7 \times 7 \times 1$ Monkhorst-Pack k-point mesh for one-unit-cell of $Cr_2S_3$. A vacuum layer of more than 15 Å was inserted along the out-of-plane direction to avoid the interactions between periodic images simulated with supercells. The out-of-plane polarization was calculated by the dipole moment correction method. To account for the strong correlation effect involving the localized 3d electron, an effective Hubbard parameter ($U_{eff} = U - J = 1.5$ eV) was applied to Cr atom, where U and J represented the Coulomb repulsion and exchange parameter, respectively.

## Data availability

Numerical data underlying the figures presented in this study are provided in the Source data file. All data are available upon request from the corresponding author. Source data are provided with this paper.

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

## Acknowledgements

This work was supported by the National Natural Science Foundation of China (92164103 to J.S.; 91964203 to J.H.; 12274050 to X.J.), the National Key R&D Program of China (2021YFA1200800 to J.S.; 2018YFA0703700 to J.H.), the Fundamental Research Funds for the Central Universities (2042023kf0187 to J.S.), and the Beijing National Laboratory for Molecular Sciences (BNLMS202001 to J.S.). The authors would like to acknowledge the Center for Electron Microscopy at Wuhan University for their substantial supports to JEM-F200 and JEM-ARM200CF. The authors also thank the support of the Center for Nanoscience and Nanotechnology at Wuhan University for the RSM and XPS characterizations.

## Author contributions

J.S. and J.H. conceived and designed the project. L.S. performed the CVD growth and etching-free transfer of $Cr_2S_3$. Y.Z. did the theoretical calculations under the guidance of X.J. L.S. conducted OM, SEM, Raman, AFM, XPS, TEM characterizations, ferroelectric and ferromagnetic measurements with the assistance of R.D., H.L., W.F., J.Y., X.L., Z.L., X.W., Y.P., Y.W., H.S., L.H., and Y.J. B.X. and Y.C. conducted the polarization hysteresis loop measurements. All of the authors commented on the manuscript.

## Competing interests

The authors declare no competing interests.
