## [Peer Review File · Nature Communications]

REVIEWER COMMENTS

Reviewer #1 (Remarks to the Author):

Two-dimensional van der Waals multiferroics have emerged as an attractive building block with immense potential to provide multifunctionality in nanoelectronics. This manuscript ensures the new discovery of intrinsic ferroelectricity in layered Cr₂S₃. Definitely, it is interesting and important that stable out-of-plane polarization is experimentally observed in room-temperatures, especially in one-unit-cell sample. I have no doubt about the novelty of the manuscript and believe it nicely fits the scope and criteria of Nat. Comm.

However, before I suggest the publication of this manuscript, one major issue needs to be clarified which is the atomic structure of the claimed Cr₂S₃. I am not questioning the experimental results but the first-principles ones although authors have shown the corresponding figures in Fig. 4 which can not convince me at this moment. It should be clarified that how the calculated structure relates to the experimental results, how the polarization reverses from one direction to the other opposite one. Especially, the initial state in Fig. 4I shows different energy with the final states that puzzles me as well. I feel that the atomic structure which provides nice two-dimensional ferroelectricity is not correctly estimated.

Besides, there are some minor issues in the manuscript need to be carefully treated. For instance, in the part of method, DFT calculation part, it is written that "The Brillouin zone integration was performed using a $3 \times 3 \times 1$ Monkhorst-Pack k-point mesh" while in the ferroelectric calculation part, we see 'The Γ -centered Monkhorst-Pack k-point mesh resolution in reciprocal space was $2\pi \times 0.025 \text{ \AA}^{-1}$ for all the structures'. Do they refer to the same case? To my knowledge, Monkhorst-Pack k-point mesh is not a Γ -centered way, what does this sentence mean?

Overall, the current status of this manuscript seems not be ready for being published on Nat. Comm.

Reviewer #2 (Remarks to the Author):

The manuscript entitled "Robust multiferroic in interfacial modulation synthesized wafer-scale one-unit-cell of chromium sulfide" reports a brand-new interface-modulated strategy to grow 1-inch one-unit-cell of Cr₂S₃ on industry-compatible c-plane sapphire. This is a new progress in 2D non-layered single crystal

fabrication. The authors also reveal that the strong interaction between Cr₂S₃ and substrate induces the interlayer-sliding of intercalated Cr atoms, which breaks the space reversal symmetry and promotes the p-d orbital hybridization between S and Cr, and then results in the emergence of room-temperature ferroelectricity/multiferroic. These results enrich the 2D multiferroic materials community and provide a platform for constructing multifunctional devices.

The reviewer thinks that this work presents a new-method for wafer-scale non-layered single-crystal synthesis, room-temperature multiferroic exploration, and physical mechanism interpretation. I recommend publishing this work in Nature Communications, after addressing the following comments.

1. The authors think that the introduction of Cr changes the sapphire surface-terminated structure and increases the interfacial interaction between Cr₂S₃ and substrate, which contribute to the domain orientation control and single crystal synthesis of Cr₂S₃. If the interfacial interaction is decoupled, whether the domain orientation of Cr₂S₃ can be controlled?
2. Please give the full name of KI, when it appears first time in the manuscript. In addition, the authors chose KI as the growth accelerant of one-unit-cell of Cr₂S₃, what is the difference between KI and NaCl? As shown in the previous literatures, NaCl is commonly employed for synthesizing large-area 2D TMDCs films.
3. The author should list out the growth substrates of Cr_mX_n in Table 2, because the substrate and the interaction between Cr_mX_n and substrate should influence the magnetic measurements.
4. During the CVD process, what is the role of hydrogen? Please discuss more about this point.
5. The authors should give more details in the synthesis and magnetic exploration of chromium-based chalcogenide. The other relevant references should be added, e.g., Mater. Today 57, 66, 2022; Adv. Mater. 34, 2107512, 2022.

Reviewer #3 (Remarks to the Author):

In this manuscript, the authors reported an interfacial modulated method to synthesize wafer-scale one-unit-cell of Cr₂S₃ on c-plane sapphire. They proposed that the introduction of Cr changed the sapphire surface-terminated structure, increased the interfacial interaction between Cr₂S₃ and sapphire, induced the parallel steps formation on sapphire surface at low temperature, which contributed to the domain orientation control of Cr₂S₃. In parallel, the strong interaction between Cr₂S₃ and substrate promoted the interlayer-sliding of intercalated Cr atoms, which broke the space reversal symmetry and resulted in the generation of room-temperature ferroelectricity/multiferroic.

I think that this manuscript presents a great breakthrough in wafer-scale growth of 2D ferroelectricity/multiferroic single crystals and offers a promising avenue for constructing low-power logic and nonvolatile memory device. I recommend to publish this work in Nature Communications after minor revision.

1. The authors proposed that the sapphire surface was changed from OH-terminated to Al-terminated structure, how to confirm this structure change?
2. For the bulk and thick Cr₂S₃ nanosheets, triangular or hexagonal morphologies are frequently observed, however, in this manuscript, the obtained Cr₂S₃ are half circular, the author should provide some discussions.
3. In Figure 4d,f, the PFM phase and amplitude hysteresis loops are not symmetric regarding the zero bias, the author should offer some explanations.
4. 2D ferroelectricity has been discovered in some TMDCs (e.g. SnSe, In₂Se₃, and CuInP₂S₆), what is the superiority of Cr₂S₃?
5. The interface modulation method in the manuscript is very interesting. Can the author discuss the differences between this method and traditional methods, such as <https://doi.org/10.1063/1.3633103> and <https://doi.org/10.1364/OL.39.005184>

Reviewer #1 (Remarks to the Author):

Two-dimensional van der Waals multiferroics have emerged as an attractive building block with immense potential to provide multifunctionality in nanoelectronics. This manuscript ensures the new discovery of intrinsic ferroelectricity in layered Cr₂S₃. Definitely, it is interesting and important that stable out-of-plane polarization is experimentally observed in room-temperatures, especially in one-unit-cell sample. I have no doubt about the novelty of the manuscript and believe it nicely fit the scope and criteria of Nat. Comm.

Our response:

We are very grateful for the reviewer's positive evaluation toward the significance of our manuscript. We also appreciate the reviewer's kind suggestion and constructive comments. These issues raised by the reviewer are considered very carefully and addressed point-by-point as follows.

However, before I suggest the publication of this manuscript, one major issue needs to be clarified which is the atomic structure of the claimed Cr₂S₃. I am not questioning the experimental results but the first-principles ones although authors have shown the corresponding figures in Fig. 4 which can not convince me at this moment. It should be clarified that how the calculated structure relates to the experimental results, how the polarization reverses from one direction to the other opposite one. Especially, the initial state in Fig. 4i shows different energy with the final states that puzzles me as well. I feel that the atomic structure which provides nice two-dimensional ferroelectricity is not correctly estimated.

Our response:

We are very thankful for the reviewer's constructive comment. We agree with the reviewer that the DFT calculations are crucial for expounding the origin of ferroelectricity in one-unit-cell of Cr₂S₃. We have optimized the atomic structures to further match the experimental results and uncovered the polarization conversion mechanism. The corresponding structure model is constructed by superimposing the atomic arrangements on the cross-sectional STEM image. As shown in **Figure R1a** (or **Fig. 4k**), the blue, yellow, and orange spheres represent the Cr, S atoms in CrS₂ layers and the self-intercalated Cr atoms, respectively. In addition, the atomic ratio of Cr and S in the theoretical model, as well as the thickness of one-unit-cell is assessed to be 11:18 and 1.73 nm, respectively, consistent with the experimental results (2:3 and 1.80 nm). The theoretical spacing of (110) plane is set to be 3.04 Å, matching well with the experimental value of 3.00 Å.

The polarization reversion from one direction to the other opposite one can be explained by the intercalation-driven sliding mechanism. As revealed in **Figure R2a** (or **Supplementary Fig. 27a**), the ground state of Cr₂S₃ is constructed by AAA stacking of CrS₂, and its central inversion symmetry should not result in the spontaneous polarization. However, the strong interfacial interaction between Cr₂S₃ and sapphire surface induces an unusual sliding of self-intercalated Cr atoms and a new ABA stacking order is built. The intercalated Cr atom in the upper interlayer forms a distorted trigonal prismatic coordination with S atoms, while the intercalated Cr atom in the lower interlayer forms a distorted trigonal antiprismatic coordination with S atoms, as displayed in **Figure R1b** (or **Fig. 4l**). During the sliding of central CrS₂ layer, the coordination environment of self-intercalated Cr atoms changes accordingly. The final state is obtained as the upper self-intercalated Cr atoms form a distorted trigonal antiprismatic coordination with S atoms and the lower self-intercalated Cr atoms form a distorted trigonal prismatic coordination with S atoms. We have optimized the atomic structures and the same energies are observed for the initial and final states, as shown in **Figure R1b** (or **Fig. 4l**). The plane-averaged charge densities along *z* direction are plotted in **Figure R2b** (or **Supplementary Fig. 27b**) to clarify the charge distribution of two polarization states. For the initial state, the charge near the lower intercalated Cr atom is more than that of the upper intercalated Cr atom, whereas the final state is

opposite. The net charge between upper and lower intercalated Cr atoms results in the interfacial charge transfer and induces the spontaneous polarization. A large barrier is obtained to be 1.4 eV/cell due to the breakdown and formation of covalent bonds between intercalated Cr atoms and S atoms during the sliding process. In addition, the remanent polarization of one-unit-cell thick Cr_2S_3 is calculated to be $0.10 \mu\text{C}/\text{cm}^2$, consistent with the experimental result (Figure R2c or Supplementary Fig. 27c).

Figure R1 (or Fig. 4k,l). **a**, Atomic structure of one-unit-cell of Cr_2S_3 with intralayer-sliding of self-intercalated Cr atoms. The blue, yellow, and orange spheres represent the Cr, S atoms in CrS_2 layers and the self-intercalated Cr atoms, respectively. **b**, Energy evolution between the two opposite polarization states of one-unit-cell of Cr_2S_3 . The energy decreasing from the centrosymmetric to non-centrosymmetric structure indicates a continuous and spontaneous phase transition between these two phases.

Figure R2 (or Supplementary Fig. 27). Polarization reversion mechanism of Cr_2S_3 . **a**, Atomic structure of one-unit-cell of Cr_2S_3 without intralayer-sliding of self-intercalated Cr atoms. **b**, Plane-averaged charge densities along z direction. **c**, DFT calculated the remanent polarization of one-unit-cell thick Cr_2S_3 .

We have added some discussions regarding the polarization reversion mechanism in **page 10** and **page 11** by “...Comparing with the pristine AAA stacking of Cr_2S_3 (Supplementary Fig. 27a), the strong interfacial interaction between Cr_2S_3 and sapphire surface induces the self-intercalated Cr atoms sliding and a new ABA stacking order is built. The intercalated Cr atom in the upper interlayer forms a distorted trigonal prismatic coordination with S atoms, while the intercalated Cr atom in the lower interlayer forms a distorted trigonal antiprismatic coordination with S atoms (Fig. 4l). During the sliding of central CrS_2 layer, the coordination environment of self-intercalated Cr atoms changes accordingly. The final state is obtained as the upper self-intercalated Cr atoms form a distorted trigonal antiprismatic coordination with S atoms and the lower self-intercalated Cr atoms form a distorted trigonal prismatic coordination with S atoms. The plane-averaged charge densities along z direction are plotted in Supplementary Fig.

27b to clarify the charge distribution of two polarization states. For the initial state, the charge near the lower intercalated Cr atom is more than that of the upper intercalated Cr atom, whereas the final state is opposite. The net charge between upper and lower intercalated Cr atoms results in the interfacial charge transfer and induces the spontaneous polarization. The energy difference between polar and non-polar structure is calculated to be 1.4 eV/cell (comparable to that of Bi₆O₉ film⁴⁹), implying the relatively high stability of polar phase. In addition, the remanent polarization of one-unit-cell thick Cr₂S₃ is calculated to be 0.10 μC/cm², consistent with the experimental result (Supplementary Fig. 27c)....”.

Besides, there are some minor issues in the manuscript need to be carefully treated. For instance, in the part of method, DFT calculation part, it is written that “The Brillouin zone integration was performed using a 3 × 3 × 1 Monkhorst-Pack k-point mesh” while in the ferroelectric calculation part, we see ‘The Γ-centered Monkhorst-Pack k-point mesh resolution in reciprocal space was 2π × 0.025 Å⁻¹ for all the structures’. Do they refer to the same case? To my knowledge, Monkhorst-Pack k-point mesh is not a Γ-centered way, what does this sentence mean? Overall, the current statue of this manuscript seems not be ready for being published on Nat. Comm.

Our response:

We are very thankful for the reviewer’s constructive comment and kind suggestion. There are two methods for automatic selecting *K* points provided by VASP, Monkhorst-Pack and Gamma-centered Monkhorst-Pack grids. During the DFT calculations, 3 × 3 × 1 Monkhorst-Pack *k*-point mesh is applied. Nevertheless, in view of the odd number of *K*-points along each dimension, the Gamma point is included in the Monkhorst-Pack method, which is the same as the Gamma-centered Monkhorst-Pack grids. According to the reviewer kind suggestion, we have revised the related description in **page 16** by “...*The Brillouin zone integration was performed using a 7 × 7 × 1 Monkhorst-Pack k-point mesh for one-unit-cell of Cr₂S₃....*”.

Reviewer #2 (Remarks to the Author):

The manuscript entitled “Robust multiferroic in interfacial modulation synthesized wafer-scale one-unit-cell of chromium sulfide” reports a brand-new interface-modulated strategy to grow 1-inch one-unit-cell of Cr₂S₃ on industry-compatible c-plane sapphire. This is a new progress in 2D non-layered single crystal fabrication. The authors also reveal that the strong interaction between Cr₂S₃ and substrate induces the interlayer-sliding of intercalated Cr atoms, which breaks the space reversal symmetry and promotes the p-d orbital hybridization between S and Cr, and then results in the emergence of room-temperature ferroelectricity/multiferroic. These results enrich the 2D multiferroic materials community and provide a platform for constructing multifunctional devices.

The reviewer thinks that this work presents a new-method for wafer-scale non-layered single-crystal synthesis, room-temperature multiferroic exploration, and physical mechanism interpretation. I recommend publishing this work in Nature Communications, after addressing the following comments.

Our response:

We are very grateful for the reviewer’s positive evaluation toward the significance of our manuscript. We also appreciate the reviewer’s kind suggestion and constructive comments. These issues raised by the reviewer are considered very carefully and addressed point-by-point as follows.

1. The authors think that the introduction of Cr changes the sapphire surface-terminated structure and increases the interfacial interaction between Cr_2S_3 and substrate, which contribute to the domain orientation control and single crystal synthesis of Cr_2S_3 . If the interfacial interaction is decoupled, whether the domain orientation of Cr_2S_3 can be controlled?

Our response:

We are very thankful for the reviewer's constructive comment. To remove or reduce the interfacial interaction between Cr_2S_3 and sapphire, $\text{Cr}_2\text{S}_3/\text{WS}_2$ vertical heterostructures are synthesized on c -plane sapphire using a two-step CVD method, and the interfacial interaction is thus decoupled by the intercalated monolayer WS_2 . As a result, Cr_2S_3 nanosheets with random orientations are observed on monolayer WS_2 , as shown in **Figure R3** (or **Supplementary Fig. 9**) different from the unidirectionally aligned Cr_2S_3 on c -plane sapphire, indicating that the domain orientation is controlled by the interfacial interaction between Cr_2S_3 and substrate. We have added some discussions in **page 4** by "...To further confirm the strong interfacial interaction determining the domain orientation of Cr_2S_3 , $\text{Cr}_2\text{S}_3/\text{WS}_2$ vertical heterostructures are synthesized on c -plane sapphire and the obtained Cr_2S_3 nanosheets possess random orientations (Supplementary Fig. 9)....".

Figure R3 (or Supplementary Fig. 9). OM image of as-grown $\text{Cr}_2\text{S}_3/\text{WS}_2$ vertical heterostructures on c -plane sapphire, showing the random orientations of Cr_2S_3 nanosheets.

2. Please give the full name of KI, when it appears first time in the manuscript. In addition, the authors chose KI as the growth accelerant of one-unit-cell of Cr_2S_3 , what is the difference between KI and NaCl? As shown in the previous literatures, NaCl is commonly employed for synthesizing large-area 2D TMDCs films.

Our response:

We are very thankful for the reviewer's constructive comment and kind suggestion. We have added the full name of KI in **page 14**. We have synthesized unidirectionally aligned Cr_2S_3 nanosheets on c -plane sapphire using NaCl as the growth accelerant, with the results shown in **Figure R4**. Nevertheless, the obtained Cr_2S_3 nanosheets are seriously etched by Cr atoms due to the fast evaporation rate of Cr with the assistance of NaCl. Therefore, NaCl is not suitable for synthesizing high-quality atomically thin Cr_2S_3 .

Figure R4. OM images of as-grown Cr_2S_3 nanosheets on c -plane sapphire using NaCl as the growth accelerant. The obtained Cr_2S_3 nanosheets are seriously etched by Cr atoms.

3. The author should list out the growth substrates of Cr_mX_n in Table 2, because the substrate and the interaction between Cr_mX_n and substrate should influence the magnetic measurements.

Our response:

We are very thankful for the reviewer's kind suggestion. We have added the growth substrates of Cr_mX_n in **Table 2**.

4. During the CVD process, what is the role of hydrogen? Please discuss more about this point.

Our response:

We are very thankful for the reviewer's constructive comment. During the CVD growth process, the introduction of hydrogen reduces the oxygen-containing impurity, decreases the nucleation density, and increases the domain size of Cr_2S_3 . As a comparison, the small and thick Cr_2S_3 nanosheets are synthesized on *c*-plane sapphire without the hydrogen assistance, as shown in **Figure R5**.

*Figure R5. OM image of as-grown Cr_2S_3 nanosheets synthesized on *c*-plane sapphire without the hydrogen assistance, showing the small domain size and high thickness.*

5. The authors should give more details in the synthesis and magnetic exploration of chromium-based chalcogenide. The other relevant references should be added, e.g., *Mater. Today* 57, 66, 2022; *Adv. Mater.* 34, 2107512, 2022.

Our response:

We are very thankful for the reviewer's kind suggestion. We have added the detailed descriptions regarding the synthesis and magnetic exploration of Cr_mX_n in **page 14** and **page 15** by “...*Before heating, 500 standard cubic centimeters (sccm) argon (Ar) was purged into the chamber for 10 minutes to remove the residual air and humidity. Subsequently, the first and second zones were heated to 170 and 980 °C, respectively, with 110 sccm Ar and 10 sccm hydrogen (H₂) as the carrier gases. The growth time was set to be 25 minutes. After completing the CVD growth process, the furnace cover was opened and cooled down to room-temperature....*” and in **page 15** by “...*Low-temperature quantum transport measurements of Cr_2S_3 were conducted in a 9T-Physical Property Measurement System (PPMS, Quantum Design, Dynacool) by constructing a four-terminated Hall bar device. The Hall resistances were measured with the perpendicular magnetic field up to 9 T, and the testing temperature range was set to be 2 to 250 K with a current of 10 μA....*”.

We have added new references in **page 2** (in **Refs. 29,30**).

Reviewer #3 (Remarks to the Author):

In this manuscript, the authors reported an interfacial modulated method to synthesize wafer-scale one-unit-cell of Cr₂S₃ on c-plane sapphire. They proposed that the introduction of Cr changed the sapphire surface-terminated structure, increased the interfacial interaction between Cr₂S₃ and sapphire, induced the parallel steps formation on sapphire surface at low temperature, which contributed to the domain orientation control of Cr₂S₃. In parallel, the strong interaction between Cr₂S₃ and substrate promoted the interlayer-sliding of intercalated Cr atoms, which broke the space reversal symmetry and resulted in the generation of room-temperature ferroelectricity/multiferroic.

I think that this manuscript presents a great breakthrough in wafer-scale growth of 2D ferroelectricity/multiferroic single crystals and offers a promising avenue for constructing low-power logic and nonvolatile memory device. I recommend to publish this work in Nature Communications after mirror revision.

Our response:

We are very grateful for the reviewer's positive evaluation toward the significance of our manuscript. We also appreciate the reviewer's kind suggestion and constructive comments. These issues raised by the reviewer are considered very carefully and addressed point-by-point as follows.

1. The authors proposed that the sapphire surface was changed from OH-terminated to Al-terminated structure, how to confirm this structure change?

Our response:

We are very thankful for the reviewer's constructive comment. The structure change of sapphire surface is convinced by multiscale characterizations. For example, the surface terminated Al atoms are directly observed by the atomic-resolution cross-sectional STEM image in **Supplementary Fig. 12**. Two XPS characteristic peaks corresponding to Al-OH (531.5 eV) and Al-O-Al (530.5 eV) are detected on the sapphire surface after annealing without the presence of Cr powders, which is different from the sapphire surface after annealing with the presence of Cr powders, where only one characteristic peak (530.5 eV) is observed (**Fig. 2f**). Such results indicate that sapphire surface terminated structure is changed by Cr atoms. Besides, the contact angle discrepancy obtained on sapphire surfaces after annealing with and without the presence of Cr powders reconfirms the structure change of sapphire surface (**Fig. 2d,e**).

2. For the bulk and thick Cr₂S₃ nanosheets, triangular or hexagonal morphologies are frequently observed, however, in this manuscript, the obtained Cr₂S₃ are half circular, the author should provide some discussions.

Our response:

We are very thankful for the reviewer's constructive comment. We agree with the reviewer that the triangular and hexagonal Cr₂S₃ nanosheets are frequently observed on mica substrates (*Adv. Mater.* **32**, 1905896 (2020)). However, in view of the different substrate and growth mechanism, the half circular Cr₂S₃ nanosheets are obtained on c-plane sapphire. The evolution of atomically thin Cr₂S₃ on c-plane sapphire obeys the step-edge-guided mechanism and the long domain edge of Cr₂S₃ is aligned with the step edge direction of sapphire. Therefore, the half circular morphology of Cr₂S₃ nanosheets is attributed to the high energy barrier of passing over the step edge. The similar phenomena are also demonstrated in CVD synthesis of h-BN (*Nature* **570**, 91–95 (2019)) and WSe₂ (*ACS Nano* **9**, 8368–8375 (2015)). We have added some discussions in **page 4** by “...*The half circular Cr₂S₃ nanosheets are observed on c-plane sapphire possibly due to the high energy barrier of passing over the step edge of sapphire....*”.

3. In Figure 4d,f, the PFM phase and amplitude hysteresis loops are not symmetric regarding the zero bias, the author should offer some explanations.

Our response:

We are very thankful for the reviewer's constructive comment. The interfacial charge and internal electric field induced by the distinct electrodes (PFM tip and Au) should result in the asymmetry of PFM phase and amplitude hysteresis loops. The similar phenomenon is also observed in other 2D ferroelectric materials, such as In_2Se_3 (*Nano Lett.* **23**, 3098–3105 (2023)). We have added some discussions in **page 9** by “...*Notably, the interfacial charge and internal electric field induced by the distinct electrodes (PFM tip and Au) result in the asymmetry of PFM phase and amplitude hysteresis loops....*”.

4. 2D ferroelectricity has been discovered in some TMDCs (e.g. SnSe, In_2Se_3 , and CuInP_2S_6), what is the superiority of Cr_2S_3 ?

Our response:

We are very thankful for the reviewer's constructive comment. We agree with the reviewer that 2D ferroelectricity has been discovered in some TMDCs (e.g. SnSe, In_2Se_3 , and CuInP_2S_6). However, most of these researches are concentrated on layered van der Waals ferroelectric materials, which suffers from the inferior stability and limited species diversity. Therefore, the emergence of room-temperature ferroelectricity in non-layered Cr_2S_3 expands the scope of ferroelectric. In addition, Cr_2S_3 possess the highest remanent polarization ($32 \mu\text{C}/\text{cm}^2$) among TMDCs, which provides a promising avenue to construct low-power nonvolatile memory devices. Besides, the robust magnetoelectric coupling is uncovered, which makes 1-inch one-unit-cell of Cr_2S_3 the largest and thinnest multiferroics.

5. The interface modulation method in the manuscript is very interesting. Can the author discuss the differences between this method and traditional methods, such as <https://doi.org/10.1063/1.3633103> and <https://doi.org/10.1364/OL.39.005184>.

Our response:

We are very thankful for the reviewer's constructive comment. The interfacial modulation method has been proposed to improve the quality of HgCdTe and enhance the infrared detector performances (*Appl. Phys. Lett.* **99**, 091101 (2011); *Opt. Lett.* **39**, 5184–5187 (2014)). However, the traditional interfacial modulation method mainly focuses on the target materials (e.g. HgCdTe), in this manuscript, the sapphire surface terminated structure is changed by Cr atoms, which enhances the interfacial interaction between Cr_2S_3 and sapphire substrate, and then determines the unidirectional growth of Cr_2S_3 and the intralayer-sliding of self-intercalated Cr atoms. We have added some discussions in **page 5** “...*The interfacial modulation method has been proposed to improve the quality of target materials (e.g. HgCdTe)^{38,39}, which provides a new direction to understand the growth mechanism of well-aligned Cr_2S_3 on c-plane sapphire....*”.

We have added these new references in **page 5** (in **Ref. 38,39**).

REVIEWER COMMENTS

Reviewer #1 (Remarks to the Author):

The manuscript has been quite improved and I am fine with the updated theoretical results.

However, a significant point still puzzles me which is the amplitude of the polarization. As it is reported, the experimental value is at the level of $1\mu\text{C}/\text{cm}^2$ which agrees the theoretical calculations. This value become $32\mu\text{C}/\text{cm}^2$ in 45nm sample that surprises me a lot since it can not be explained by the calculations.

In the calculation, a periodic model is used implying the absence of the thickness and surface effect. I believe it nicely shows what happened in the 2nm sample.

If the authors would like to emphasize the large polarization in thicker samples, I think it is necessary to tell readers what happens when the thickness of samples increases.

Reviewer #2 (Remarks to the Author):

The authors have addressed all the reviewers' concerns and this manuscript can be published in Nature Communications. Particularly, the authors have provided additionally experimental and theoretical results to further clarify the interfacial modulated growth mechanism and the origin of ferroelectric polarization in atomically thin Cr₂S₃. Therefore, I recommend to publish this work without further revision.

Reviewer #3 (Remarks to the Author):

What are the noteworthy results?

Answer: The authors provide a interface modulated strategy to grow 1-inch one-unit-cell of non-layered chromium sulfide with unidirectional orientation on industry-compatible c-plane sapphire.

Will the work be of significance to the field and related fields? How does it compare to the established literature? If the work is not original, please provide relevant references.

Answer: Yes, this work is significance to the field and related fields. This work has realized one-unit-cell of multiferroic materials design and wafer-scale synthesis compare to the established literature.

Does the work support the conclusions and claims, or is additional evidence needed?

Answer: The work has supported the conclusions and claims and isn't need additional evidence.

Are there any flaws in the data analysis, interpretation and conclusions? Do these prohibit publication or require revision?

Answer: The data analysis, interpretation and conclusions are consistent with the experimental results. I am agree to publish the paper in Nature Communications.

Is the methodology sound? Does the work meet the expected standards in your field?

Answer: The methodology is sound and the work meet the expected standards.

Is there enough detail provided in the methods for the work to be reproduced?

Answer: It provides enough detail in the methods for the work to be reproduced.

Reviewer #1 (Remarks to the Author):

The manuscript has been quite improved and I am fine with the updated theoretical results. However, a significant point still puzzles me which is the amplitude of the polarization. As it is reported, the experimental value is at the level of $1 \mu\text{C}/\text{cm}^2$ which agrees the theoretical calculations. This value become $32 \mu\text{C}/\text{cm}^2$ in 45 nm sample that surprises me a lot since it can not be explained by the calculations. In the calculation, a periodic model is used implying the absence of the thickness and surface effect. I believe it nicely shows what happened in the 2 nm sample. If the authors would like to emphasize the large polarization in thicker samples, I think it is necessary to tell readers what happens when the thickness of samples increases.

Our response:

We are very glad to know that the supplementary theoretical results are helpful to understand the origin of ferroelectricity in Cr_2S_3 . We are also thankful for the reviewer's new comment regarding the remanent polarization of thick Cr_2S_3 .

The strong interfacial interaction between Cr_2S_3 and sapphire induces the sliding of self-intercalated Cr atoms and CrS_2 layers between the intercalated Cr atoms, which breaks the space reversal symmetry and results in the emergence of ferroelectricity. As the thickness increases, the interfacial interaction and the intralayer-sliding of self-intercalated Cr atoms are weakened accordingly, as confirmed by the cross-section STEM images in **Figure 4i – j**. Even so, the influence of depolarization field on the ferroelectricity of thick Cr_2S_3 is also reduced. In addition, the remanent polarization value of thick Cr_2S_3 is the summation of different unit-cells along z direction, therefore, a large remanent polarization is obtained in the 45 nm sample comparing with one-unit-cell of Cr_2S_3 . Notably, the similar phenomena are also demonstrated in other ferroelectric materials, such as $\text{PbZr}_{0.2}\text{Ti}_{0.8}\text{O}_3$ films (*Nat. Commun.* **8**, 15549 (2017)) and 3R- MoS_2 nanosheets (*Nat. Commun.* **13**, 7696 (2022)).

According to the reviewer's kind suggestion, we have added some discussions in **page 10** by “...*The cumulative effect of self-intercalated Cr atoms and CrS_2 layers sliding, as well as the weakened depolarization field should result in the enhancement of remanent polarization in thick Cr_2S_3 ^{42,43}*”.

We have added new references in **page 10** (in **Ref. 42,43**).

Reviewer #2 (Remarks to the Author):

The authors have addressed all the reviewers' concerns and this manuscript can be published in Nature Communications. Particularly, the authors have provided additionally experimental and theoretical results to further clarify the interfacial modulated growth mechanism and the origin of ferroelectric polarization in atomically thin Cr_2S_3 . Therefore, I recommend to publish this work without further revision.

Our response:

We are very thankful for the reviewer's recommendation. Again, we would like to thank the reviewer's thoughtful comments and suggestion, which we think have helped to greatly improve the readability and clarity of our manuscript.

Reviewer #3 (Remarks to the Author):

What are the noteworthy results?

Answer: The authors provide an interface modulated strategy to grow 1-inch one-unit-cell of non-layered chromium sulfide with unidirectional orientation on industry-compatible c-plane sapphire.

Will the work be of significance to the field and related fields? How does it compare to the established literature?

If the work is not original, please provide relevant references.

Answer: Yes, this work is significance to the field and related fields. This work has realized one-unit-cell of multiferroic materials design and wafer-scale synthesis compare to the established literature.

Does the work support the conclusions and claims, or is additional evidence needed?

Answer: The work has supported the conclusions and claims and isn't need additional evidence.

Are there any flaws in the data analysis, interpretation and conclusions? Do these prohibit publication or require revision?

Answer: The data analysis, interpretation and conclusions are consistent with the experimental results. I am agree to publish the paper in Nature Communications.

Is the methodology sound? Does the work meet the expected standards in your field?

Answer: The methodology is sound and the work meet the expected standards.

Is there enough detail provided in the methods for the work to be reproduced?

Answer: It provides enough detail in the methods for the work to be reproduced.

Our response:

We are very thankful for the reviewer's recommendation. Again, we would like to thank the reviewer's thoughtful comments and suggestion, which we think have helped to greatly improve the readability and clarity of our manuscript.

REVIEWER COMMENTS

Reviewer #1 (Remarks to the Author):

I feel sorry that the current response does not fully convince me. Actually, it is unlikely to have such large amplitude of out-of-polarization when it has negligible ionic polar displacement along this direction. Usually, interfacial interaction may affect but with small amplitude. Besides, both the up and down polarization are significantly enhanced which puzzles me as well. Therefore I still feel this part is not solid enough.

To be honest, if the interaction with substrate can make such significant modification, that would be a big news in this field. I kindly suggest authors to confirm these results carefully since such big number will definitely attract plenty of attention from people working on ferroelectricity.

I am ok with the rest part of the results which are enough for a nice publication on Nat. Communications.

If authors disagree to make further change, I would like to let Editor to make the final decision.

Reviewer #1 (Remarks to the Author):

I feel sorry that the current response does not fully convince me. Actually, it is unlikely to have such large amplitude of out-of-polarization when it has negligible ionic polar displacement along this direction. Usually, interfacial interaction may affect but with small amplitude. Besides, both the up and down polarization are significantly enhanced which puzzles me as well. Therefore, I still feel this part is not solid enough. To be honest, if the interaction with substrate can make such significant modification, that would be a big news in this field. I kindly suggest authors to confirm these results carefully since such big number will definitely attract plenty of attention from people working on ferroelectricity. I am ok with the rest part of the results which are enough for a nice publication on Nat. Communications. If authors disagree to make further change, I would like to let Editor to make the final decision.

Our response:

We are very thankful for the reviewer's constructive comment. We agree with the reviewer that the interfacial interaction is not the only factor that determines such a large amplitude of out-of-plane polarization in Cr₂S₃. The ionic polar displacement and interlayer charge transfer possibly influence the amplitude, and related theoretical explorations are expected to be made in the future to further clarify the origin of ferroelectricity. According to the reviewer's kind suggestion, we have replaced the polarization hysteresis loop of 45.0 nm-thick Cr₂S₃ in **Figure 4f** with a thin sample of 13.0 nm and the corresponding remanent polarization value is calculated to be 4.30 μC/cm². Even so, this value is still higher than other 2D ferroelectric materials, as shown in **Table 1**. We have also revised the relevant discussions in **page 10** by "...*The remanent polarization value as large as 4.30 μC/cm² is observed for the Cr₂S₃ nanosheet with the thickness of 13.0 nm (Fig. 4f), which is higher than other 2D ferroelectric materials (Table 1). Besides of the interfacial interaction, the ionic polar displacement and interlayer charge transfer possibly contribute to enhance the polarization of thick Cr₂S₃, and related theoretical explorations are expected to be made in the future. Furthermore, the cumulative effect of self-intercalated Cr atoms and CrS₂ layers sliding, as well as the weakened depolarization field should also result in the polarization elevation, as have been demonstrated in other ferroelectric materials^{42,43}. In addition, the macroscopic ferroelectric hysteresis loop measurement of bulk Cr₂S₃ is performed, with the result shown in Supplementary Fig. 25....*".

Thanks again for the reviewer's comment regarding the remanent polarization of Cr₂S₃, which is very helpful for understanding the origin of ferroelectricity.

REVIEWERS' COMMENTS

Reviewer #1 (Remarks to the Author):

The updated results looks more reasonable and make me feel better. At this moment, I would kind suggest authors to check any unexpected typos, word's spellings Besides, I think the manuscript seems ok for the publication on Nat. Comm.

Reviewer #1 (Remarks to the Author):

The updated results looks more reasonable and make me feel better. At this moment, I would kind suggest authors to check any unexpected typos, word's spellings Besides, I think the manuscript seems ok for the publication on Nat. Comm.

Our response:

We would like to thank you for reviewing our paper, we appreciate your insightful comments on our research. We have carefully checked all the grammar in our manuscript according to your kind suggestion.

Again, we would like to thank the other reviewers' comments and suggestion, which we think have helped to greatly improve the readability and clarity of our manuscript.